# Fugitive emissions of natural gas in York, United Kingdom: Adapting

## 2 existing algorithms parameters to be based on instrument

## 3 specifications.

- 4 Thomas C. Moore<sup>1</sup>, James R. Hopkins<sup>1,2</sup>, Will S. Drysdale<sup>1,2</sup>, Stuart Young<sup>1</sup>, Sri Hapsari Budisulistiorini<sup>1</sup>,
- 5 Marvin D. Shaw<sup>1,2</sup>, James L. France<sup>3,4</sup>, David Lowry<sup>3</sup> and James D. Lee<sup>1,2</sup>
- 6 Wolfson Atmospheric Chemistry Laboratories, University of York, York YO10 5DD, United Kingdom
- <sup>2</sup>National Centre for Atmospheric Science, University of York, York YO10 5DD, United Kingdom
- 8 <sup>3</sup>Royal Holloway, University of London, Earth Sciences, Egham, United Kingdom
- <sup>4</sup>Environmental Defence Fund Europe, Avenue des Arts 47-49, Brussels, Belgium

Correspondence to: james.lee@york.ac.uk

#### Abstract

10 11 12

13 14

25

- Reducing methane emissions has become increasingly important in recent years due to its importance for radiative forcing. Of
- the many sources of methane, fugitive emissions from a country's domestic natural gas network are one that can have a direct
- impact on the citizens of a country. Previous studies have shown the ability to detect these emissions by use of mobile surveys
- measuring methane, some of these use secondary co-emitted compounds as a means of confirming the nature of the emission.
- This study aims to adapt existing algorithms parameters by investigating the limitations of equipment used within the platform
- used for mobile surveys. This has led to reduced enhancement parameters as well as reduced time clustering parameters. These
- changes suggest that previous methods may underpredict the number of Leak Indications (LIs) by 53.5% with number of LIs
- detected in the old method being 27 and the new method detecting 58. When source appointment was included as a core step
- within the algorithm itself, the total fugitive natural gas emissions within a city was reduced from 185.10 L min<sup>-1</sup> to 60.23 L
- 24 min<sup>-1</sup>, nearly three times lower.

## 1 Introduction

- Following COP26 and the methane pledge (European Commission and United States of America 2021), methane and its
- emissions have received increased attention. The pledge states that the signatories will attempt to reduce their methane
- emissions by 30% of their 2020 levels by 2030 and was brought about due to increasing concern over the potency of methane
- as a greenhouse gas. With a warming potential 28 times greater than CO<sub>2</sub> over a 100-year timescale and 84 times higher over

## https://doi.org/10.5194/egusphere-2025-5348 Preprint. Discussion started: 27 November 2025 © Author(s) 2025. CC BY 4.0 License.

- a 20-year timescale (IPCC, 2021). Although the growth rate of methane concentrations in the atmosphere has slowed since the
- 1980s, it is still responsible for 20% of the warming caused by long-lived greenhouse gases since pre-industrial times (Kirschke
- et al., 2013).
- The difficulty in reducing methane emissions is in part due to the wide range of emission sources, both natural and
- anthropogenic. 60% of methane emissions are thought to be anthropogenic in nature (Karakurt et al., 2013, Saunois et al.,
- 2020) and of those emissions, by far the largest contribution arises from the agricultural sector with the two main areas of this
- being enteric fermentation from ruminant livestock like cattle and emissions from paddy fields (Karakurt et al., 2013, Yusuf
- et al., 2012). Although there are means of reducing the emissions from these areas, such as changing the feedstock of the cattle,
- for example grass silage to maize silage (Bačėninaitė et al., 2022), there is still a requirement for social change and land
- management change, both of which are difficult to implement for a 2030 deadline.
- After agriculture the largest contribution to anthropogenic emissions is from the energy sector with oil, natural gas and coal
- having relatively similar contributions to methane emissions. Natural gas is of particular importance to the UK, with it being
- the 19th largest country emitter of methane from the natural gas network (Scarpelli et al., 2022).
- One of the sources of methane emissions from the natural gas network is fugitive emissions. A fugitive emission is an
- unexpected or unwanted emission of gas from a pressurised network that is not detected by standard means (Sotoodeh, 2021).
- Within the natural gas network, they are commonly referred to colloquially as "Gas Leaks", however the stigma surrounding
- this term, both from industrial operators and the public, means the term fugitive emission is preferable to be used where
- possible.
- In the United Kingdom in 2023, 63.5 billion cubic metres of natural gas was consumed (Energy Institute, 2024). This is used
- in a range of applications including industrial use, electricity generation and domestic use. Of the UKs natural gas
- consumption, 33.8 % is from the domestic sector (DESNZ, 2024), with 73.8% of households in England and Wales using
- mains gas for either heating or cooking purposes. (Stewart et al., 2024), in 2022 it was estimated that 117 kT of methane as a
- result of fugitive emissions related to natural gas distribution (NAEI, UK Emissions Data Selector).
- The natural gas network consists of production or importing of natural gas; transport of high-pressure natural gas through the
- National Gas' National Transmission System (NTS), a network of over 5,000 miles of high-pressure steel pipes and more than
- 500 above ground installations; and finally, distribution of natural gas to homes and businesses which occurs through the 4
- Gas Distribution Networks (GDNs) in the UK.
- A GDN first reduces the pressure from the NTS before then supplying natural gas to its customers. The GDN responsible for
- York covers 2.7 million homes and businesses not just across Yorkshire, but also the northeast of England and northern
- Cumbria, which means the operation of tens of thousands of kilometres of pipework. This all leads to many unknowns in the
- locations of fugitive emissions. To combat this, many previous studies have implemented mobile measurement approaches
- centred around the detection of areas with elevated methane.

## 1.1 Previous Mobile Measurement Methodology

utilise detection of an enhancement in CH<sub>4</sub> concentrations, the major component of natural gas. These algorithms define an enhancement based on whether CH<sub>4</sub> mixing ratios are higher than a certain value (Phillips et al., 2016), are above a certain 65 66 percentile in measured readings (Hopkins et al., 2016, Chamberlain et al., 2016) or by using an outlier detection model (Keyes et al., 2020). 67 68 The paper upon which our methodology is based defines Observed Peaks (OPs) as methane enhancements > 110% of a 2.5-69 minute rolling background of the mean CH<sub>4</sub> concentrations two minutes before and after each measured point, and that they 70 occur over less than 160 m (von Fischer et al., 2017). Enhancements occurring within 5 seconds of each other are then grouped 71 together. It is required that drives are repeated multiple times and Leak Indications (LIs) are determined after grouping OPs 72 that occur within 20 m of one another and seeing which of these grouped clusters contain OPs from separate drives. The LIs 73 are then quantified into emission rates in L min<sup>-1</sup>, using an equation derived from the results from a controlled release

There have been many different approaches to designing an algorithm to detect fugitive emissions of natural gas, all of which

- $(release\ rate) = 0.1178 + 0.08267 \times M 0.005175 \times A + 0.08626 \times K$
- Where:

experiment.

88

- M is the maximum CH<sub>4</sub> reading
- A is the ppm.metres index
- K is the ratio of ppm.metres to maximum CH<sub>4</sub>
- This methodology was then further developed in (Weller et al., 2019), this changed the baseline such that it was the median
- CH<sub>4</sub> value over 2.5 minutes, the spatial grouping of OPs to LIs from 20 m to 30 m and the quantification equation to:
- $ln (excess CH_4) = -0.988 + 0.817 \times ln (emission rate)$
- In (Maazallahi et al., 2020), it was proposed that the existing methodology categorised certain burning emissions as fugitive
- emissions, to counter this an additional stage using CO<sub>2</sub> ratios with CH<sub>4</sub> was introduced to filter burning emissions.
- Source attribution was also used in (Fernandez et al., 2022), using isotopic measurements of CH<sub>4</sub> in addition to ethane:
- methane ratios. Variations of this algorithm has been used in many major cities across the US and Canada (Ars et al., 2020;
- Weller et al., 2022) as well as European cities (Defratyka et al., 2021; Fernandez et al., 2022; Wietzel et al., 2023; Vogel et

al., 2024). This paper attempts to detect smaller enhancements of methane by adapting the parameters to be specific to the

- limitations of the instrumentation used. The paper also explores the effect of introducing a source attribution filter at the OP
- stage of the algorithm and how this affects the number and the magnitude of LIs.

## 2 Methodology

#### 2.1 Instrumentation

The WACL Air Sampling Platform (WASP) detailed in (Wagner et al., 2021) is the base for these measurements, the inlet for the WASP's sampling is located at the front of the van on the driver's side, meaning that the vehicle will sample the centre of the road regardless of direction of travel. The WASP has since been updated to include a Quark-Elec QK-AS07-0183 for GPS readings. For the measurements surrounding natural gas, the WASP was equipped with a Los Gatos Microportable Greenhouse Gas Analyser (MGGA) for measurements of methane and CO<sub>2</sub>, Iterative Cavity Enhanced Differential Optical Absorption Spectrometer (ICAD) for measurements of NO<sub>x</sub> (NO<sub>2</sub> + NO), and an Aerodyne Tuneable Infrared Laser Direct Absorption Spectrometer (TILDAS) Laser Trace Gas Analyser for measurements of ethane (Yacovitch et al., 2014). Measurements of ethane are calibrated using a three point calibration of a high standard of ethane (17.5 ppb), medium (2.5 ppb) and a zero, where calibration standard concentrations were confirmed via GC-MS. For each drive a calibration is performed before and after the drive itself, a linear regression is performed to find the slope and intercept of the calibration concentrations versus measured concentrations. The average is taken for the two calibrations to account for instrument drift during the sampling drive and the resulting equation is then used to apply a correction to ethane concentrations.

 $C_2H_{6\ corrected} = C_2H_{6\ uncorrected}\ .\ m+c$ 

Where:

- m = Gradient of calibration concentration vs mean response averaged over the two calibrations
- c = Intercept of calibration concentration vs mean response averaged over the two calibrations

## 2.1.1 Instrument Response Time

Response time of the MGGA is reported as 

2 separate valve setups for a total of 20 repeats of low-high-low transitions in the concentration of ethane. The transition times were located by eye and then the transition time to go from 90% of the maximum value to 10% of the maximum value was calculated (Symonds, 2017). The transition time on the first valve ranged from 0.7 - 1.1 s with a mean value of 0.9 s, the second valve having responses ranging from 0.7 - 1.4 s also with a mean response of 0.9 s, giving confidence in a sub 1 s response rate from the TILDAS and therefore showing the capability of a sub 1 s response in both instruments. The data however was still averaged to 1 s with a 1 s clustering time due to the data now being limited by the data acquisition rate of the WASP's GPS.

Figure 1: Example response transition of TILDAS high concentration to low concentration, normalised to maximum recorded response

https://doi.org/10.5194/egusphere-2025-5348 Preprint. Discussion started: 27 November 2025 © Author(s) 2025. CC BY 4.0 License.

#### 2.1.2 Variation in methane measurements

For better understanding of minimum detectable enhancements of methane possible with the equipment available a variance experiment using compressed air flowing through the instrument for 2 hours. The standard deviation of methane measurements over this period was calculated to inform the enhancement criteria for the methane detection algorithm.

For 2 hours compressed air was flown through the Los Gatos MGGA, measured at a median value of 7.2 ppm, the standard deviation of these results was found to be 0.006 ppm. The enhancement criteria was calculated as 5 times this standard

deviation of these results was found to be 0.006 ppm. The enhancement criteria was calculated as 5 times this standard deviation divided by the median baseline, in this case 1.005 times the baseline. However, this assumes a stable baseline that is replicated in the field. In reality, when applying this enhancement criteria it leads to the detection of enhancements that are too small to be reliably quantified. Instead, the variation in the calculated baseline for drives was found and it was therefore

determined that an enhancement of 1.05 times the baseline should be used instead.

## 2.2 Driving Route

York is a city in the north-east of England with a population over 200,000. A driving campaign took place over two separate weeks in May and June of 2024 resulting in 18 drives of a "flower petal" route staying within the outer ring roads of the A64 and A1237 and mainly sampling residential areas of the city, the majority of the roads sampled on the route were only driven in one direction but due to the position of the sampling inlet this allowed the middle of the road to be the sampled point. The route was driven this number of times due to findings that to capture > 90 % of emissions, a route should be driven at least 5 – 8 times over separate days (*Luetschwager et al.*, 2021). The route was chosen as it covers multiple different neighbourhoods within York, but was not intended to be used to compare measurements to the cities emissions inventory due to only covering a small fraction of the total miles of road within the York urban area, 27 miles of a total 507 administered by the local authority (*Department for Transport*, 2025).

Figure 2: Map of the route taken in WASP surveys.

#### 2.3 Enhancement Detection Algorithm

Unlike the previous iterations of the algorithm, OPs are clustered within 1 s as opposed to 5 s. With a faster instrument response, it was expected that the measurements would more readily distinguish between two separate enhancements that occurred spatially close to one another. Additionally, by clustering over a time of 5s, assuming an average speed of 20 miles per hour (8.9 m/s), this would mean the potential to cluster together enhancements 44.5m apart, whereas a cluster of 1s would at most be clustering enhancements 8.9 m apart, the reason for this change was discussed in 2.1.1. Enhancement criteria was also changed such that instead of an enhancement needing to be more than 110% of the baseline value to instead being an enhancement related to the results of the variance experiment to allow detection of smaller enhancements, this variation is discussed in 2.1.2. LI determination occurred after identifying the source type of each OP, to allow LI analysis to occur on

## https://doi.org/10.5194/egusphere-2025-5348 Preprint. Discussion started: 27 November 2025 © Author(s) 2025. CC BY 4.0 License.

177178

OPs of the same source type, to further reduce the chance of comparing long standing thermogenic fugitive emissions with possible nearby single occurrence pyrogenic and biogenic emissions.

## 2.4 Controlled Release Experiment

To attain a quantification equation specific to the equipment used at York, a controlled release experiment was conducted at the Bedford Aerodrome. The controlled release took place at the Bedford Aerodrome over 5 days in June of 2024. A MiniCRF was deployed to manage releases of methane and ammonia, while a MidiCRF was deployed for releases of ethane. In total there were 41 releases lasting an average of 30 minutes each. Releases consisted of varying amounts of methane (0.2 – 70.48 L min<sup>-1</sup>), ethane  $(0-7.01 \text{ L min}^{-1})$  and ammonia  $(0-7.87 \text{ L min}^{-1})$  to reflect a range of methane sources, including natural gas and farm emissions. Releases were from a mixture of linear vertical releases, a muti emission point ring, multi point source emissions and single point releases, occurring at heights ranging from ground level to 3m elevation. Over the course of the experiment wind speeds were measured using four Gill Met Pak Pro instruments deployed at 3,6,9 and 12 metres elevation, winds were recorded as 1 minute vector averages. Average wind speed over the 5 days was 3.87 m s<sup>-1</sup> with wind speeds ranging from 0 - 9.75 m s<sup>-1</sup>. During each release, an initial period was spent locating the plume before sampling the plume at set distances for 10 repeats before stepping further away in distance for another set of 10 repeats, repeated until the plume was either lost, or a lack of driveable ground was left available, it was noted that larger releases were detectable further away. However, as the data from the controlled release was intended to be used in quantifying sub-road and near-road fugitive emissions of natural gas, a maximum distance of 30m from the controlled release point was applied due to this reflecting the maximum road widths typically found within a city like York (Essex Planning Officers Association, 2018). Of the 41 releases conducted in the controlled release, only 26 releases were able to be used for processing data, this is due to several reasons, including some releases not having detectable enhancements. Within these 26 releases, 3525 methane enhancements were detected over distances between 5.8 m and 382.1 m from the release point, the majority of releases detected further away from the release point were from higher emission rate releases. When this was filtered to include enhancements from only 30 m from the release point this resulted in including 1226 enhancements from 23 drives.

Figure 3: Density plot of number of detected enhancements against distance from release point

Figure 4: Density plot of number of detected enhancements against distance from release point (Limited to 0-30 m)

## 2.4.1 Quantification equation

A quantification equation to calculate emission rate from CH<sub>4</sub> measurements could be calculated using enhancements that were detected within 30 m of the controlled release emission point, to better reflect the enhancements likely to be detected from under-road natural gas emissions. By first using the methane detection algorithm a linear regression model could then be used to find a relationship between the release rate and the maximum enhancement. 1226 enhancements were detected from 23 separate releases that were within 30 m of the emission point. These enhancements were determined from CH<sub>4</sub> releases ranging from 0.49 to 70.48 L min<sup>-1</sup>. A linear regression was then performed using the equation:

 $ln (maxexcess CH_4) = b_0 + b_1 * ln (emission rate) + \varepsilon, \varepsilon \sim^{iid} N(0, \sigma^2)$ 

*Figure 5:* Plot of the ln of the maximum detected  $CH_4$  enhancement against the ln of the known  $CH_4$  release rate.

This resulted in the quantification equation;

 $ln (maxexcess CH_4) = -1.2668 + 0.6323 * ln (emission rate)$ 

## 2.4.2 Instrument Lag Time

For each of the releases, the lag time between detecting an ethane enhancement and a methane enhancement was calculated. With knowledge of the response times of the instruments it was expected that the TILDAS would respond to an enhancement before the MGGA, however this assumes that both instruments receive the same packet of air at the same time, while, in reality, the packet of air will take a different amount of time to flow through manifold to each instrument. To find this more accurate lag time of the instruments, the maximum methane enhancement for each pass was found, following this the maximum ethane enhancement was found (that occurred within 5 seconds of the methane). The 5 second window was selected as on transects of the controlled release the van travelled at roughly 20 mph. Over the course of 10 s (5s either side of the methane maximum) this would mean 85 m covered in the van, the average length of a transect being 180 m. The time lag between ethane and methane showed that in most cases (88.1%), maximum ethane concentration preceded maximum methane concentration with a mean lag of 2.7 s before and a median of 3.8 s before. Observing a window of max methane to 5 s before max methane resulted in a mean lag of 3.3 s from ethane to methane and a median lag of 3.9 s. This helped inform the detection algorithm to look for maximum ethane within a window only up to 5s before the maximum methane.

223224

Figure 6: Density plot of time lag of the maximum ethane measurement from the maximum methane measurement.

226227

Figure 7: Density plot of time lag of the maximum ethane measurement from the maximum methane measurement (Only including ethane measurements that precede the methane)

## 2.5 Source Appointment

Source determination using ethane-methane ratios has been shown to be effective, due in part to the knowledge that ethane is present in measurable quantities in thermogenic gas but not biogenic gas (Fernandez et al., 2022), ethane:methane (C<sub>2</sub>:C<sub>1</sub>)

ratios can be used in order to determine the source of a methane emission. Demonstrated in (Fernandez et al., 2022; Defratyka et al., 2021; Lowry et al., 2020; Yacovitch et al., 2014), C<sub>2</sub>:C<sub>1</sub> < 0.005 is associated with biogenic sources, >0.005 to <0.09 are thermogenic and >0.1 are considered pyrogenic or combustion. To complete this time alignment of methane and ethane values needs to be completed due to them being measured on separate instruments, the criterion for aligning these times is based on the results from 2.4.2. Additionally enhancements are removed where the R<sup>2</sup> of CO<sub>2</sub>:CH<sub>4</sub> is greater than 0.9 to ensure no combustion sources are wrongly assigned as thermogenic.

Figure 8: Relationship between  $CH_4$  and  $C_2H_6$  for three OPs of different source types located during the sampling campaign.

## 3 Results

## 3.1 Results of York Drives

17 drives were conducted across the route of York, the raw data was taken from 10Hz files for methane (MGGA) and ethane (TILDAS) and time averaged to 1Hz data to be of the same response time as the WASPs internal components (GPS). The data was then processed to remove data taken when speeds were 0 or > 40 mph as well as removing data within the area of Wolfson

Atmospheric Chemistry Laboratories due to this also being the location within which calibrations and other instrument tests were conducted. A rolling 2.5-minute median background of CH<sub>4</sub> was then taken and enhancements were determined as any CH<sub>4</sub> measurement taken that was greater than 1.05 times the calculated background, the enhanced readings were then clustered such that any elevated reading within one second of another were assumed to correspond to the same enhancement. These enhancements were then spatially averaged such that 467 OPs were detected over the course of the 18 drives.

*Figure 9:* Colour map of  $CH_4$  concentration from one of the York drives.

For each of these OPs the maximum ethane value was found from the time of maximum methane to 5 seconds prior. The two instruments' data were then aligned for each OP such that time of max methane was equal to time of max ethane. A linear regression was then taken of values from 5 seconds prior to the maximum methane to 5 seconds after and a source type was assigned such that  $C_2: C_1 < 0.005$  is associated with biogenic sources, >0.005 to <0.09 are thermogenic and >0.1 are considered

pyrogenic or combustion.

This meant that of the 467 OPs, 178 (38.1%) were found to be thermogenic in origin. Each Thermogenic OP was then clustered to find others within 30 m of one another and filtered to ensure that clusters contained OPs occurring on at least two separate

drives, to remove any OPs occurring from an event happening on only one drive. The remaining clusters were then averaged into LIs such that the CH<sub>4</sub> enhancement was the maximum from the OPs and the latitude and longitude was a weighted spatial average, resulting in 24 Thermogenic LIs from the 178 thermogenic OPs. The smallest leak rate was determined to be 0.26 L min<sup>-1</sup> and the largest being 16.04 L min<sup>-1</sup>. When the source type filter is omitted, this results in 58 LIs with leak rates ranging from 0.24 to 23.10 L min<sup>-1</sup>. In terms of cumulative leak rates, without source attribution cumulative leak rate would be 185.1 L min<sup>-1</sup> whereas when source attribution was factored in a cumulative leak rate of 60.2 L min<sup>-1</sup> was found.

## 3.1.1 Industry applicability

As many gas distribution companies have signed up to voluntary emission reporting programmes, such as the Oil and Gas Methane Partnership (OGMP) 2.0, they are now obligated to report emissions through measurement based methods. One of the most popular methods for such a reporting programme is through comprehensive, repeated vehicle based measurement surveys of an operator's gas network. Here, we have a repeated route of measurements where thermogenic emissions have been reported at certain locations throughout the campaign. It is therefore interesting from a mitigation perspective to investigate how many times each of those thermogenic emissions were identified over the course of the campaign.

Figure 10: Wind direction consistency and wind speed per Thermogenic leak indication (Labelled with number of enhancements in each LI)

The effect of wind on detection of LIs was initially investigated by calculating the mean resultant length of wind directions when a Thermogenic OP was detected. This was calculated using the equation:

$$\rho = \frac{1}{n} \sqrt{\left(\sum_{i=1}^{n} \blacksquare \cos \theta_{i}\right)^{2} + \left(\sum_{i=1}^{n} \blacksquare \sin \theta_{i}\right)^{2}}$$

Where:

- ρ is mean resultant length

- n is number of data points

-  $\theta_i$  is the angle in radians

For this analysis  $\rho$  is close to 1 when the wind directions are concentrated (similar) and close to 0 when more dispersed.

Figure 10 shows that for the majority of LIs detected in York  $\rho$  is close to 1, suggesting that most LIs occur away from the road and require correct wind direction in order to detect them. For LIs where  $\rho$  is lower it can also be noted that the mean wind speeds also tend to be lower, suggesting that these LIs may occur closer to the mobile platform therefore not needing a specific direction of wind in order to detect them.

Figure 11: Pie charts of each LI showing number of drives they were detected vs not

Number of drives is a large factor in the probability of detecting an LI. Each LI requires the enhancement to be detected on at least 2 separate drives, for the 24 LIs detected over the course of this campaign 12 LIs were detected on 2 drives, 6 were detected on 3 drives, 4 on 5 drives and 2 on 7, this meant that the average probability of detection was 0.18. This low probability of detection highlights the need for surveys with multiple passes.

#### 3.2 Pyrogenic emissions

While 178 of the 467 OPs were determined to be Thermogenic, 41 were assigned as biogenic (8.8%) and 196 were Pyrogenic (42.0%). NO<sub>x</sub>:CO<sub>2</sub> ratios were investigated for these OPs using the same methodology used for the CH<sub>4</sub>:C<sub>2</sub>H<sub>6</sub> source assignment. 112 of the 196 pyrogenic OPs were able to be analysed in this way, 85 of these 112 OPs (75.9%) had a NO<sub>x</sub>:CO<sub>2</sub> ratio < 0.88 x 10<sup>-3</sup>. This implied that the majority of pyrogenic emissions did not originate from traffic but more likely emissions from domestic heat and power generation (such as emissions from domestic boilers) (Cliff et al., 2025). When these 85 Pyrogenic boiler OPs were clustered over 30m and filtered to only include multiple drives (much like how LIs were determined from Thermogenic OPs) it resulted in only 5 recurring emissions of this type, this is to be expected, as emissions from boilers are less likely to be a consistent emission source than a fugitive emission from a natural gas pipe.

Emissions from pyrogenic sources are compared at the OP stage on a drive by drive basis due to the high unlikelihood of these emissions being detected on multiple drives.

Figure 12: Total leak rate (L min<sup>-1</sup>) contribution from each source type for each of the drives.

Thermogenic emissions from natural gas had the highest overall contribution to OPs across all drives and had the highest individual contribution on 10 of the 18 drives. However, for 7 of 18 drives pyrogenic emissions related to heating and cooking had the second highest overall contribution to methane emissions along the sampling route. For all source types, the majority of enhancements detected were of leak rates less than 5 L min<sup>-1</sup>, no non-domestic pyrogenic or biogenic emission was detected with a leak rate greater than 10 L min<sup>-1</sup>. Although a few higher leak rate outliers were detected for domestic pyrogenics, thermogenic emissions were the only source type with emissions seen consistently at leak rates up to 20 L min<sup>-1</sup>.

Figure 13: Histograms of leak rate distributions by source type.

## 3.3 Comparison to previous methods

The main alterations to this methodology from that present in *Weller et al.*, 2019 (and other studies that were based of this method) were that enhancement criteria was changed from 1.1 times the baseline to 1.05 times the baseline, clustering by time was changed such that emissions within 1 s of each other were clustered instead of those within 5s and finally that a source determination filter was applied as opposed to no source determination occurring or that this came later in the analysis. By comparing the methodology with the new one derived from the instrument limitations we find the following:

| Enhancement<br>Criteria | Time Clustering Criteria / s | Source Determination Included? | Number of OPs | Number of LIs |
|-------------------------|------------------------------|--------------------------------|---------------|---------------|
| 110% of baseline        | 5                            | No                             | 179           | 27            |
|                         |                              | Yes                            | 66            | 6             |
|                         | 1                            | No                             | 216           | 29            |
|                         |                              | Yes                            | 79            | 7             |
| 105% of baseline        | 5                            | No                             | 357           | 58            |
|                         |                              | Yes                            | 144           | 23            |
|                         | 1                            | No                             | 467           | 58            |
|                         |                              | Yes                            | 178           | 24            |

Table 1: Number of detected OPs and LIs depending on changing algorithm parameters

Showing the new methodology could locate more LIs. Additionally, when the predicted leak rates were compared from the non-source filtered LIs it showed that in addition to being able to locate more LIs, the changes to enhancement criteria and time clustering have also led to the ability to locate LIs of a greater range of leak rates. For the previous method the 27 detected LIs range in leak rate between 0.70 and 23.10 L min<sup>-1</sup>, with a mean Leak Rate of 4.53 L min<sup>-1</sup>, 75% of LIs have a leak rate between 1.71 and 4.10 L min<sup>-1</sup>, whereas with the new enhancement criteria and time clustering parameter there was a range in leak rate of 0.24 to 23.10 L min<sup>-1</sup>, a mean of 3.19 L min<sup>-1</sup> and 75% of LIs having a Leak rate between 0.51 and 2.26 L min<sup>-1</sup>. The total methane emissions from LIs were 122.43 L min<sup>-1</sup> for the old method compared to 185.10 L min<sup>-1</sup> for the new method. This however changes when the source filtering is factored in where the Leak Rates ranged from 0.26 to 16.04 L min<sup>-1</sup> with a mean leak rate of 2.51 L min<sup>-1</sup> and 75% of LIs having a leak rate between 0.38 and 2.85 L min<sup>-1</sup> additionally the cumulative emissions was 60.23 L min<sup>-1</sup>. If this source filtering is applied to the old method however we see that for the 6 LIs, leak rates ranged from 1.64 to 16.04 L min<sup>-1</sup> with a mean of 4.74 L min<sup>-1</sup> and a cumulative emission of 28.48 L min<sup>-1</sup>. Showing the old methods underprediction in total methane emissions of 33.9% when not source filtered and 59.4% when source filtering is considered.

Figure 14: Box plot showing the varying detected Leak Rates depending on methodology used

## 4. Conclusions

This study focused on using the specifications of the equipment used for the surveying to better inform a detection algorithm. Enhancement criteria was determined by investigating the variance of the MGGA, although laboratory experiments suggested the instrumentation was capable of detecting enhancements at a minimum of 1.005 times the baseline, in field experiments showed that a preferable limit was 1.05 times the baseline. Response rate of the instruments was calculated to inform the time window for clustering, with both MGGA and TILDAS having sub one second response rate, the time clustering was instead limited to one second due to the limitations of GPS data collection speed. These changes show previous methodologies would result in detection of 33.9% lower cumulative emissions.

Source appointment proved to be a useful tool for predicting emissions directly related to natural gas, when source filtering was introduced at the OP stage of detection, it resulted in only 41.4% of LIs still being detected as opposed to the non-source filtered method. Additionally, this meant a reduction of 67.5% in the total estimated emissions of methane.

https://doi.org/10.5194/egusphere-2025-5348 Preprint. Discussion started: 27 November 2025 © Author(s) 2025. CC BY 4.0 License.

Additionally source filtering has helped to highlight that although Thermogenic emissions from natural gas are the highest contributor to methane emissions, pyrogenic emissions related to domestic heat and power generation also provide a high but often overlooked contribution to a cities methane emissions. This new method has shown that with changing enhancement criteria and time clustering parameters, it is able to detect many more LIs and a wider range of leak rates and that by applying a source type filter at the OP detection stage it is capable of reducing the overprediction of methane emissions from natural gas. However, the methodology has ability to improve further, primarily by using instrumentation that is capable of detecting methane and ethane on one instrument, so as to remove uncertainty related to time lag between the two instruments, but secondly by having all instrumentation and hardware able to operate at a sub one second time rate in order to reduce the time clustering parameter limit. Code / Data availability Code and data will be made available upon request. **Author Contribution** Contributed to conception: TM, JH, WD, JL. Contributed to data acquisition: TM, JH, WD, SY, SHB, MS, JL. Contributed to analysis and interpretation of data: TM, JH, WD, SY, JF, JL. Initial draft of paper: TM. Subsequent drafts and/or revisions to paper: TM, JH, WD, SY, DL, JF, JL. Approved the submitted version of this paper for publication: TM, JH, WD, SY, SHB, MS, JF, DL, JL. **Competing Interests** The authors declare that they have no conflict of interest. Acknowledgements We would like to thank the INGENIOUS (UnderstandING the sourcEs, traNsformations and fates of IndOor air pollUtantS) project, NERC grant number NE/W002256/1, for providing access to their data in the early stages of the method development. Additionally we would like to thank both the National Physical Laboratory (NPL) and the MOMENTUM (Mobile Observations and quantification of Methane Emissions to inform National Targeting, Upscaling and Mitigation) project, NERC

grant number NE/X014649/1, for organising and providing access to the controlled release experiment.

2024), 2023.

References 382 Ars, S., Vogel, F., Arrowsmith, C., Heerah, S., Knuckey, E., Lavoie, J., Lee, C., Pak, N.M., Phillips, J.L. and Wunch, D., 383 Investigation of the spatial distribution of methane sources in the greater Toronto area using mobile gas monitoring systems. 384 Environ. Sci. Technol., 54(24), pp.15671-15679. https://doi.org/10.1021/acs.est.0c05386, 2020. 385 386 Bačeninaitė, D., Džermeikaitė, K. and Antanaitis, R., Global warming and dairy cattle: How to control and reduce methane emission. Animals, 12(19), p.2687. https://doi.org/10.3390/ani12192687, 2022. 387 388 389 Chamberlain, S.D., Ingraffea, A.R. and Sparks, J.P., Sourcing methane and carbon dioxide emissions from a small city: 390 Influence of natural gas leakage and combustion. Environ. Pollut., 218, pp.102-110, 391 https://doi.org/10.1016/j.envpol.2016.08.036, 2016. 392 393 Cheng J, Schloerke B, Karambelkar B, Xie Y, Aden-Buie G. leaflet: Create Interactive Web Maps with the JavaScript 394 'Leaflet' Library. R package version 2.2.3.9000, https://rstudio.github.io/leaflet/, 2025 395 396 Cliff, S.J., Drysdale, W., Lewis, A.C., Møller, S.J., Helfter, C., Metzger, S., Liddard, R., Nemitz, E., Barlow, J.F. and Lee, J. 397 D., Evidence of Heating-Dominated Urban NO<sub>x</sub> Emissions. Environ. Sci. Technol. 59(9), pp.4399-4408. 398 https://doi.org/10.1021/acs.est.4c13276, 2025 399 400 Defratyka, S.M., Paris, J.D., Yver-Kwok, C., Fernandez, J.M., Korben, P. and Bousquet, P., Mapping urban methane sources 401 in Paris, France. Environ. Sci. Technol., 55(13), pp.8583-8591. https://doi.org/10.1021/acs.est.1c00859, 2021 402 403 Department for Energy Security and Net Zero (DESNZ), Energy Trends: Natural Gas, Energy Trends September 2024, 404 https://assets.publishing.service.gov.uk/media/66f423473b919067bb48270e/Energy Trends September 2024.pdf (accessed 405 December 2024), 2024 406 Department for Transport: Road Length Statistics, RDL0102: Road length (miles) by road type and local authority in Great 407 408 Britain, https://www.gov.uk/government/statistical-data-sets/road-length-statistics-rdl (accessed April 2025), 2025 409 410 Energy Institute, Statistical Review of World Energy, Natural gas consumption in the United Kingdom (UK) from 2003 to 411 2023 (in billion cubic meters), 412 https://www.energyinst.org/ data/assets/pdf file/0004/1055542/EI Stat Review PDF single 3.pdf (accessed December

417

420

424

428

434

437

440

- Essex Planning Officers Association, The Essex Design Guide, Design Details, 2018 Edition, V3,
- https://www.essexdesignguide.co.uk/media/2402/design-details-v3.pdf (Accessed December 2024), 2018.
- European Commission, United States of America, Global methane pledge,
- <a href="https://www.ccacoalition.org/sites/default/files/resources//Global%20Methane%20Pledge.pdf">https://www.ccacoalition.org/sites/default/files/resources//Global%20Methane%20Pledge.pdf</a> (accessed July 2025), 2021
- Fernandez, J.M., Maazallahi, H., France, J.L., Menoud, M., Corbu, M., Ardelean, M., Calcan, A., Townsend-Small, A., van
- der Veen, C., Fisher, R.E. and Lowry, D., Street-level methane emissions of Bucharest, Romania and the dominance of
- urban wastewater. Atmos. Environ-X, 13, p.100153. <a href="https://doi.org/10.1016/j.aeaoa.2022.100153">https://doi.org/10.1016/j.aeaoa.2022.100153</a>, 2022.
- Hopkins, F. M.; Kort, E. A.; Bush, S. E.; Ehleringer, J. R.; Lai, C. T.; Blake, D. R.; Randerson, J. T. Spatial patterns and
- source attribution of urban methane in the Los Angeles Basin. J. Geophys. Res.: Atmos. 121 (5), 2490–2507,
- https://doi.org/10.1002/2015JD024429, 2016.
- IPCC, 2021: Climate Change 2021: The Physical Science Basis. Contribution of Working Group I to the Sixth Assessment
- Report of the Intergovernmental Panel on Climate Change Masson-Delmotte, V., P. Zhai, A. Pirani, S.L. Connors, C. Péan,
- S. Berger, N. Caud, Y. Chen, L. Goldfarb, M.I. Gomis, M. Huang, K. Leitzell, E. Lonnoy, J.B.R. Matthews, T.K. Maycock,
- 432 T. Waterfield, O. Yelekçi, R. Yu, and B. Zhou (eds.)]. Cambridge University Press, Cambridge, United Kingdom and New
- York, NY, USA, In press, https://doi.org/10.1017/9781009157896., 2021.
- Karakurt, I., Aydin, G. and Aydiner, K., Sources and mitigation of methane emissions by sectors: A critical review. Renew.
- Energ., 39(1), pp.40-48. https://doi.org/10.1016/j.renene.2011.09.006, 2012.
- Keyes, T., Ridge, G., Klein, M., Phillips, N., Ackley, R. and Yang, Y., An enhanced procedure for urban mobile methane
- leak detection. Heliyon, 6(10). https://doi.org/10.1016/j.heliyon.2020.e04876, 2020.
- Kirschke, S., Bousquet, P., Ciais, P., Saunois, M., Canadell, J.G., Dlugokencky, E.J., Bergamaschi, P., Bergmann, D., Blake,
- D.R., Bruhwiler, L. and Cameron-Smith, P., Three decades of global methane sources and sinks. Nat. Geosci., 6(10), pp.813-
- 823. https://doi.org/10.1038/ngeo1955, 2013.
- Lowry, D., Fisher, R. E., France, J. L., Coleman, M., Lanoisellé, M., Zazzeri, G., Nisbet, E. G., Shaw, J. T., Allen, G., Pitt,
- 446 J., and Ward, R. S.: Environmental baseline monitoring for shale gas development in the UK: Identification and geochemical

- characterisation of local source emissions of methane to atmosphere, Sci. Total Environ., 708, 134600,
- https://doi.org/10.1016/j.scitotenv.2019.134600, 2020

- Luetschwager, E., von Fischer, J.C. and Weller, Z.D., Characterizing detection probabilities of advanced mobile leak
- surveys: Implications for sampling effort and leak size estimation in natural gas distribution systems. Elem. Sci. Anth., 9(1),
- p.00143. https://doi.org/10.1525/elementa.2020.00143, 2021.

- Maazallahi, H., Fernandez, J.M., Menoud, M., Zavala-Araiza, D., Weller, Z.D., Schwietzke, S., Von Fischer, J.C., Denier
- Van Der Gon, H. and Röckmann, T., Methane mapping, emission quantification, and attribution in two European cities:
- Utrecht (NL) and Hamburg (DE). Atmos. Chem. Phys., 20(23), pp.14717-14740. https://doi.org/10.5194/acp-20-14717-
- **2020**, 2020.

- National Atmospheric Emissions Inventory (NAEI), UK Emissions Data Selector,
- <a href="https://naei.energysecurity.gov.uk/data/data-selector">https://naei.energysecurity.gov.uk/data/data-selector</a>. Selected emissions data for the year 2022, methane emissions related
- to gas leakage from gas distribution 1B2b5. (accessed June 2025)

- Phillips, N.G., Ackley, R., Crosson, E.R., Down, A., Hutyra, L.R., Brondfield, M., Karr, J.D., Zhao, K. and Jackson, R.B.,
- Mapping urban pipeline leaks: Methane leaks across Boston. Environ. Pollut., 173, pp.1-4.
- <u>https://doi.org/10.1016/j.envpol.2012.11.003</u>, 2013.

- Saunois, M., Stavert, A.R., Poulter, B., Bousquet, P., Canadell, J.G., Jackson, R.B., Raymond, P.A., Dlugokencky, E.J.,
- Houweling, S., Patra, P.K. and Ciais, P., The global methane budget 2000–2017. Earth Syst. Sci. Data, pp.1-136.
- https://doi.org/10.5194/essd-12-1561-2020, 2019.

- Scarpelli, T.R., Jacob, D.J., Grossman, S., Lu, X., Qu, Z., Sulprizio, M.P., Zhang, Y., Reuland, F., Gordon, D. and Worden,
- 472 J.R., Updated Global Fuel Exploitation Inventory (GFEI) for methane emissions from the oil, gas, and coal sectors:
- evaluation with inversions of atmospheric methane observations. Atmos. Chem. Phys., 22(5), pp.3235-3249.
- https://doi.org/10.5194/acp-22-3235-2022, 2022.

- Sotoodeh, K., Why packing adjustment cannot stop leakage: Case study of a ball valve failing to seal after packing
- adjustment during fugitive emission as per ISO 15848–1. Eng. Fail. Anal., 130, p.105751.
- <u>https://doi.org/10.1016/j.engfailanal.2021.105751</u>, 2021.

486

490

494

499

502

506

- Stewart I., Bolton P.; Households off the gas-grid and prices for alternative fuels; House of Commons Library,
- https://researchbriefings.files.parliament.uk/documents/CBP-9838/CBP-9838.pdf (accessed December 2024), 2024.
- Symonds, J, August 15 2017, On Instrument Time Response: What it means, what it isn't, and why it matters, [Article],
- LinkedIn. <a href="https://www.linkedin.com/pulse/instrument-time-response-what-means-why-matters-jonathan-symonds/">https://www.linkedin.com/pulse/instrument-time-response-what-means-why-matters-jonathan-symonds/</a>
- (Accessed November 2024)
- Vogel, F., Ars, S., Wunch, D., Lavoie, J., Gillespie, L., Maazallahi, H., Röckmann, T., Nęcki, J., Bartyzel, J., Jagoda, P. and
- Lowry, D., Ground-Based Mobile Measurements to Track Urban Methane Emissions from Natural Gas in 12 Cities across
- Eight Countries. Environ. Sci. Technol., 58(5), pp.2271-2281. https://doi.org/10.1021/acs.est.3c03160, 2024.
- von Fischer, J.C., Cooley, D., Chamberlain, S., Gaylord, A., Griebenow, C.J., Hamburg, S.P., Salo, J., Schumacher, R.,
- Theobald, D. and Ham, J., Rapid, vehicle-based identification of location and magnitude of urban natural gas pipeline
- leaks. Environ. Sci. Technol., 51(7), pp.4091-4099. https://doi.org/10.1021/acs.est.6b06095, 2017.
- Wagner, R. L., Farren, N. J., Davison, J., Young, S., Hopkins, J. R., Lewis, A. C., Carslaw, D. C., and Shaw, M. D.:
- Application of a mobile laboratory using a selected-ion flow-tube mass spectrometer (SIFT-MS) for characterisation of
- volatile organic compounds and atmospheric trace gases, Atmos. Meas. Tech., 14, 6083–6100, https://doi.org/10.5194/amt-
- 14-6083-2021, https://doi.org/10.5194/amt-14-6083-2021, 2021.
- Weller, Z.D., Yang, D.K. and von Fischer, J.C., An open source algorithm to detect natural gas leaks from mobile methane
- survey data. PLoS One, 14(2), p.e0212287. https://doi.org/10.1371/journal.pone.0212287, 2019.
- Weller, Z.D., Im, S., Palacios, V., Stuchiner, E. and von Fischer, J.C., Environmental injustices of leaks from urban natural
- gas distribution systems: patterns among and within 13 US metro areas Environ. Sci. Technol., 56(12), pp.8599-8609.
- https://doi.org/10.1021/acs.est.2c00097, 2022.
- Wietzel, J.B. and Schmidt, M., Methane emission mapping and quantification in two medium-sized cities in Germany:
- Heidelberg and Schwetzingen. Atmos. Environ-X, 20, p.100228. https://doi.org/10.1016/j.aeaoa.2023.100228, 2023.
- Yacovitch, T.I., Herndon, S.C., Roscioli, J.R., Floerchinger, C., McGovern, R.M., Agnese, M., Pétron, G., Kofler, J.,
- Sweeney, C., Karion, A. and Conley, S.A., Demonstration of an ethane spectrometer for methane source
- identification. Environ. Sci. Technol., 48(14), pp.8028-8034. https://doi.org/10.1021/es501475q, 2014.

https://doi.org/10.5194/egusphere-2025-5348 Preprint. Discussion started: 27 November 2025 © Author(s) 2025. CC BY 4.0 License.

- Yusuf, R.O., Noor, Z.Z., Abba, A.H., Hassan, M.A.A. and Din, M.F.M., Methane emission by sectors: a comprehensive
- review of emission sources and mitigation methods. Renewable and Sustainable Energy Reviews, 16(7), pp.5059-5070.
- https://doi.org/10.1016/j.rser.2012.04.008, 2012.