# Peer review of "Fugitive emissions of natural gas in York, United Kingdom: Adapting"

_EGUsphere, 2025_

## Referee Comment (RC3)

I read the manuscript titled "Fugitive emissions of natural gas in York, United Kingdom: Adapting existing algorithms parameters to be based on instrument specifications" with a great interest and motivation. This work is of high relevance to current mitigation actions.

This manuscript presents a methodologically focused study that adapts mobile methane survey algorithms to better reflect instrument limitations, with application to the detection of natural gas fugitive emissions in York, UK. The study includes controlled release experiments to derive a quantification equation, use of mobile measurements as tracers for attribution and evaluates instrument functionality.

**Overall assessment:**

The work has high importance for the benefit of emission mitigation, industrial application and policy making. However, there are significant concerns regarding clarity, presentation of results, and statistical treatment. Several sections contain incomplete sentences and formatting errors. The quantification equation derivation is inadequately explained, and key results are buried in lengthy, poorly formatted description and unrest arrangement and information. The manuscript would benefit from a thorough revision to improve readability, clarify methodological steps, and better highlight the scientific novelty. Authors can enhance scientific rigour of the work ay applying intercomparison of emission rates derived from similar equations presented in previous studies (see comments below) and discuss why emission rates outcomes might be statistically similar or different based on same methane enhancement signals.

**Detailed Comments**

Below please find comments by sections, lines, figures, equations and tables.

*Sections*

**Section 1.1**

Good initial summary of previous algorithms, but this section can be improved, but the equations from von Fischer et al. (2017) and Weller et al. (2019) are presented without clear explanation of terms (e.g., ppm.metres index). A brief description would help non-specialist readers. Please also read and refer to other relevant studies in which they also derived similar equations, Tettenborn et al. (2025) and Joo et al. (2024). Authors can make an intercomparison emission rate quantifications derived from these equations in a table and discuss the results.

**Sections 2.1.1 and 2.1.2**

Are these efforts made to address instruments drifts? If so, explanations and descriptions can be provided in a better format.

**Section 2.2**

It appears that the measurements only took place on the main roads? Why didn't you choose the residential alleys and avenues? In urban surveys those narrower streets have high significance.

**Section 2.4.1**

This is an important part of your work and should be discussed and described in more details. See comments about the intercomparison with other equations introduced previously.

**Section 2.4.2**

This section including the graphs can be placed in the supplementary information and main results can be used in the algorithm in the mani manuscript.

**Section 2.5**

The attribution part using mobile ethane and carbon dioxide is of high relevant. Are the values in X-axis are methane readings (including the background) or the values show methane enhancements? This is important because it is unclear how you classified the values around the 2 ppm into the source specific categories? For further use of ethane and carbon dioxide, I refer you to Fig. S12 (See SI from Maazallahi et al. (2020)) and Sect. S3 and Sect. S5 from Maazallahi et al. (2023).

**Section 3.2**

Do you intend to only discuss the pyrogenic sources here? I can see from Fig. 12 that other sources are also shown in the figure.

**Section 3.3**

Please see comments with regards to the use of other equations. An intercomparison can be made here and discussed in details. This can be a very significant part of your work.

*Lines*

**L1-3:** Title is clear but could be more concise. Consider: "Adapting mobile methane survey algorithms to instrument specifications: a case study in York, UK".

**L19-20:** "This study aims to adapt existing algorithms parameters…". grammatically awkward. Suggest: "This study adapts existing algorithm parameters by investigating the limitations of the mobile survey platform instrumentation."

**L21:** The claim that old methods may underpredict LIs by 53.5% is striking but should be contextualized: is this due to sensitivity changes or source filtering?

**L23:** Emission reduction from 185.10 to 60.23 L min$^{-1}$ is very significant.

**L28-30:** This sentence reads awkward.

**L45-61:** The transition to UK natural gas network is relevant, but the paragraph could better link to the study's objectives. Better to reduce non-necessary information and provide information about emissions.

**L59:** The sentence on "many unknowns" is vague. Consider rephrasing to emphasize the need for improved detection methods.

**L76–85:** The introduction of source attribution ($CO_2$, $C_2H_6$ and isotopes) is well-placed and sets up the study's contribution.

**L86–90:** The final paragraph clearly states the paper's aims. Well done.

**L91–105:** Instrument details are thorough, but the ethane calibration formula is presented without defining "mean response." The calibration procedure can be moved to supplementary information.

**L112–119:** The description of TILDAS valve modifications is technical and unclear. However, the rationale for aiming for "true 10Hz" vs. 5Hz is not fully justified. It is also unclear why other parameters are important in terms of emission rate quantifications. These descriptions belong to supplementary information and main findings can be placed in the mani manuscript.

- Why did you the measurements in normalized format?
- Better show the whole measurement graphs.

**L160-162:** Here you point out an important relation between two parameters; driving speed vs sampling frequency. With lower speed and higher measurement frequency, there is higher possibility to capture methane enhancement signals.

**L169-170:** do you suggest that a quantification equation should be introduced for each city and instrument used for individual survey? This means that in every urban surveys (city and instrument dependency) a set of control release experiments should be performed.

*Equations*

Equations are not numbered throughout the manuscript.

**L283 – L208 – L202 – L106 etc.:** equations are not numbered.

**L283:** what are those back boxes in the equation?

*Figures*

Figures are not referenced within the text. Many figures can be moved to the supplementary information and main figures can be kept in the main manuscript.

**Figure 10 –** Use of wind direction and LI assignment are very unclear. It appears that the wind speed values are not corrected and include driving speed as well. The figure is very vague and values are questionable.

**Figure 11:** texts are in small fronts. What are the use of those LI numbers? How do you interpret those numbers with regards to e.g. Fig. 10. It is better to discuss specific LIs in details.

*Tables*

**Table 1 –** Results provided in this table require further, detailed and clear explanation and discussion in the main text. Results from other algorithms can be also included in the table.

**Recommendation**

**Major revisions required.** The core science is understandable, but the manuscript requires significant restructuring, clarification of methods, correction of formatting errors, and improved data presentation before it can be considered for publication.

**References**

Tettenborn, J., Zavala-Araiza, D., Stroeken, D., Maazallahi, H., van der Veen, C., Hensen, A., Velzeboer, I., van den Bulk, P., Vogel, F., Gillespie, L., Ars, S., France, J., Lowry, D., Fisher, R., and Röckmann, T.: Improving Consistency in Methane Emission Quantification from the Natural Gas Distribution System across Measurement Devices, EGUsphere [preprint], https://doi.org/10.5194/egusphere-2024-3620, 2025.

Joo, J., Jeong, S., Shin, J., Chang, D. Y.: Missing methane emissions from urban sewer networks, Environmental Pollution, Volume 342, 123101, ISSN 0269-7491, https://doi.org/10.1016/j.envpol.2023.123101, 2024.

Maazallahi, H., Fernandez, J. M., Menoud, M., Zavala-Araiza, D., Weller, Z. D., Schwietzke, S., von Fischer, J. C., Denier van der Gon, H., and Röckmann, T.: Methane mapping, emission quantification, and attribution in two European cities: Utrecht (NL) and Hamburg (DE), Atmos. Chem. Phys., 20, 14717–14740, https://doi.org/10.5194/acp-20-14717-2020, 2020.

Maazallahi, H., Delre, A., Scheutz, C., Fredenslund, A. M., Schwietzke, S., Denier van der Gon, H., and Röckmann, T.: Intercomparison of detection and quantification methods for methane emissions from the natural gas distribution network in Hamburg, Germany, Atmos. Meas. Tech., 16, 5051–5073, https://doi.org/10.5194/amt-16-5051-2023, 2023.

von Fischer, J. C., Cooley, D., Chamberlain, S., Gaylord, A., Griebenow, C. J., Hamburg, S. P., Salo, J., Schumacher, R., Theobald, D., and Ham, J.: Rapid, Vehicle-Based Identification of Location and Magnitude of Urban Natural Gas Pipeline Leaks, Environ. Sci. Technol., 51, 4091–4099, https://doi.org/10.1021/acs.est.6b06095, 2017.

Weller, Z. D., Yang, D. K., and von Fischer, J. C.: An open source algorithm to detect natural gas leaks from mobile methane survey data, PLoS One, 14, e0212287, https://doi.org/10.1371/journal.pone.0212287, 2019.